# SCas4D: Structural Cascaded Optimization for Boosting Persistent 4D Novel View Synthesis

**Jipeng Lyu**                                                                    *lvjipenglv@gmail.com*
*University of Illinois Urbana-Champaign*

**Jiahua Dong**                                                                   *jiahuad2@illinois.edu*
*University of Illinois Urbana-Champaign*

**Yu-Xiong Wang**                                                                 *yxw@illinois.edu*
*University of Illinois Urbana-Champaign*

**Reviewed on OpenReview:** *https://openreview.net/forum?id=YkycjbKjYP*

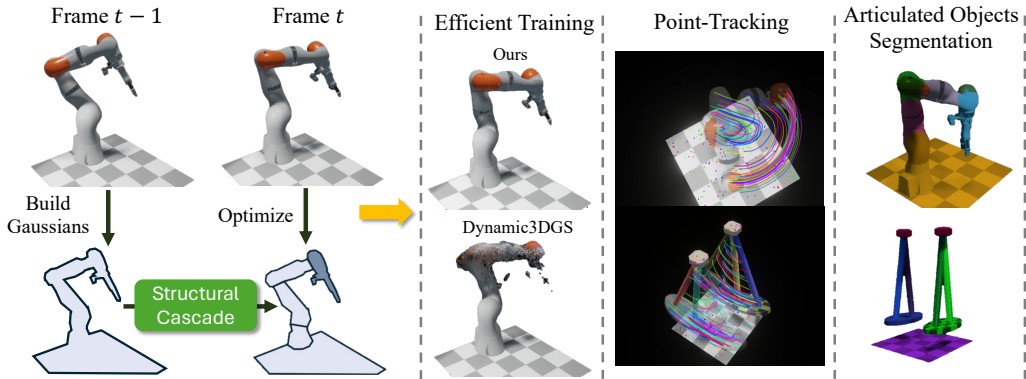

Figure 1: Our method achieves satisfying rendering results with 100 training iterations per frame. Leveraging learned deformation information, we also demonstrate successful articulated object segmentation.

## Abstract

Persistent dynamic scene modeling for tracking and novel-view synthesis remains challenging, particularly due to the complexity of capturing accurate deformations while maintaining computational efficiency. In this paper, we present SCas4D , a novel cascaded optimization framework that leverages inherent structural patterns in 3D Gaussian Splatting (3DGS) for dynamic scenes. Our key insight is that real-world deformations often exhibit hierarchical patterns, where groups of Gaussians undergo similar transformations. By employing a structural cascaded optimization approach that progressively refines deformations from coarse part-level to fine point-level adjustments, SCas4D achieves convergence within 100 iterations per time frame while maintaining competitive quality to the state-of-the-art method with only 1/20th of the training iterations. We further demonstrate our method's effectiveness in self-supervised articulated object segmentation, establishing a natural capability from our representation. Extensive experiments demonstrate our method's effectiveness in novel view synthesis and dense point tracking tasks. Please find our project page at https://github-tree-0.github.io/SCas4D-project-page/.

# 1 Introduction

Dynamic novel-view synthesis provides a powerful framework for modeling dynamic 3D scenes, with applications in fields such as AR/VR, robotics, and autonomous driving. The ability to learn and render dynamic scenes can enable immersive, interactive experiences. Recent advances (Chen et al., 2022; 2023a; Fridovich-Keil et al., 2022; Hu et al., 2023; Müller et al., 2022; Chen et al., 2023b; Garbin et al., 2021; Hedman et al., 2021; Reiser et al., 2023; Wizadwongsa et al., 2021; Chen et al., 2021; Niemeyer et al., 2022; Wynn & Turmukhambetov, 2023; Yu et al., 2021), inspired by Neural Radiance Field (NeRF) (Mildenhall et al., 2020), have leveraged radiance fields for 3D scene modeling. However, the inherent limitations of NeRF, including its high computational demand for network queries and volume rendering. Moreover, its implicit representation restricts some downstream applications, such as precise tracking and articulated object segmentation.

The recent development of 3D Gaussian Splatting (3DGS)(Kerbl et al., 2023) has notably improved efficiency in rendering static scenes by representing the 3D world with a collection of Gaussians and employing efficient rasterization. This insight has led to explorations of dynamic scene rendering with 3DGS(Wu et al., 2024; Duan et al., 2024; Sun et al., 2024; Luiten et al., 2024), with some methods (Luiten et al., 2024; Abou-Chakra et al., 2024; Zhang et al., 2022) focusing on learning transformations (e.g., position and rotation changes) for each Gaussian between frames. These methods achieve realistic deformation tracking over time. However, the structural information of 3DGS is underexplored. Specifically, the motion in 3D scenes often follows different structural patterns, such as rigid parts, non-rigid parts, and static backgrounds. While SCGS Huang et al. (2024) tries to apply sparse control points to represent motion, they fail to capture accurate and dense point-level tracking. Their requirements for correctly distributed control points further limit the application.

Inspired by these observations, we would like to answer the question that *"Can 4D scenes directly benefit from the structural information of vanilla 3DGS."* Since objects in real-world scenes often consist of multiple parts that exhibit similar deformations, we propose a structural cascaded optimization approach that organizes the Gaussians in a top-down manner. In the coarse-level, we optimize the 3D parts to approximate their deformation in the new time frame. Following this, the fine-level optimization will further improve the deformation of each Gaussian. To ensure appropriate scale change between different levels, we adopt optimization with three levels that balance efficiency with the ability to capture fine-grained motion.

Without significant modifications to 3DGS, our methods show the great potential of the underlying structural information of 3DGS. By the structural cascaded optimization, we achieve a *20× speedup* over Dynamic3DGS (Luiten et al., 2024) in training time. In the meanwhile, we maintain the ability to deliver comparable tracking performance for dense points and further provide the capability of articulated object segmentation, as shown in Figure 1. Extensive experiments demonstrate the effectiveness of our method in different tasks.

In summary, our contributions can be concluded as:

- We explore the possibility of utilizing internal structural information from 3DGS for dynamic scenes, significantly accelerating the convergence speed.

- We introduce a cascaded structural optimization strategy with a multi-level deformation function that captures rotation, translation, and scaling. An articulated object segmentation method is proposed for 3DGS.

- We achieve highly competitive performance in novel view rendering and point-tracking. Our method also shows the ability of high-quality articulated object segmentation.

# 2 Related Works

**Static Novel-View Synthesis** has become popular in 3D vision in recent years. Specifically, given a set of images from different camera poses, high-fidelity rendered images on novel views are expected. The potential of achieving photorealistic results on this task is revealed by Neural Radiance Field (NeRF) (Mildenhall

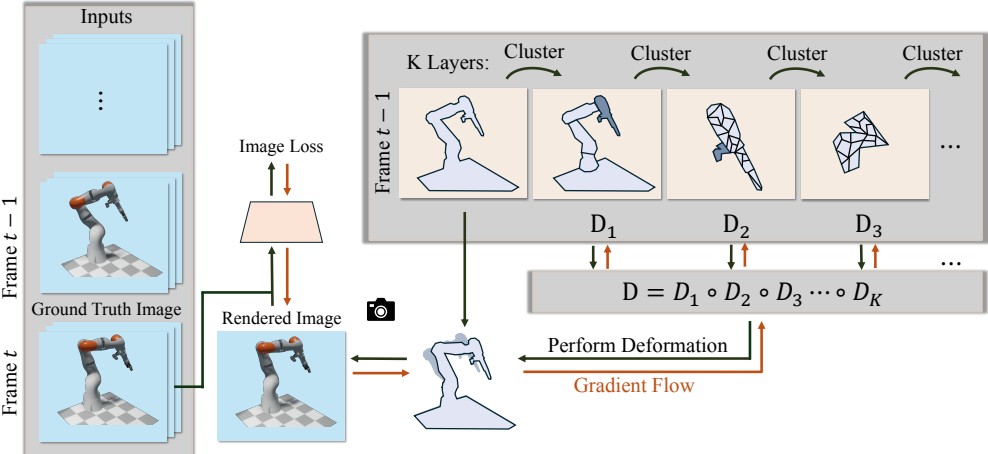

Figure 2: Our method first utilizes the Gaussians from the previous frame $t-1$ and the new inputs for frame $t$ to learn the deformation $D$ between these two frames. These Gaussians are organized into cascaded clusters with $K$ layers. For each cluster layer, we learn a deformation function. Finally, the deformation $D$ of each Gaussian is obtained by nesting these deformation functions.

et al., 2020), which encodes the scene as a fully connected deep network. Following this, a series of works are proposed to improve the efficiency, rendering quality, storage consumption, and other aspects of NeRF (Chen et al., 2022; 2023a; Fridovich-Keil et al., 2022; Hu et al., 2023; Müller et al., 2022; Chen et al., 2023b; Garbin et al., 2021; Hedman et al., 2021; Reiser et al., 2023; Wizadwongsa et al., 2021; Chen et al., 2021; Niemeyer et al., 2022; Wynn & Turmukhambetov, 2023; Yu et al., 2021). However, the design of costly volume rendering and neural networks makes the improvements very challenging, especially in balancing the time efficiency and rendering quality. Recently, 3D Gaussian Splatting (Kerbl et al., 2023) is proposed to elegantly solve this problem by explicit 3D Gaussian representation and differentiable rasterization.

Our work is highly inspired by this but extends from static scenes to dynamic scenes. In particular, we start from static 3D Gaussians and optimize towards the dynamic scene. The natural representation of 3D Gaussians allows for explicit modeling of deformation and high efficiency for both training and inference.

**Dynamic Novel-View Synthesis** is a more challenging task in dynamic scenes. Inspired by the success of NeRF (Mildenhall et al., 2020), various attempts have been made to model the dynamics (Attal et al., 2023; Cao & Johnson, 2023; Fang et al., 2022; Li et al., 2022b;c; 2021; 2023; Park et al., 2021a;b; Pumarola et al., 2021; Fridovich-Keil et al., 2023; Yang et al., 2022; Weng et al., 2022). These works solve the dynamic problem by different routes. Specifically, some works (Li et al., 2022b; Weng et al., 2022; Yang et al., 2022; Zhao et al., 2022) focus on certain scenarios like human motion and leverage prior knowledge, such as human skeletons, to facilitate the synthesis. While achieving impressive results, the modeling strategy cannot be applied to general cases. Deformation-based methods (Attal et al., 2023; Park et al., 2021a;b; Pumarola et al., 2021) build a canonical stage and warp the other frames to this stage. This approach can be applied to more general scenes but can't work well on complex scenes with high variations. Impressed by the high rendering speed of 3DGS (Kerbl et al., 2023), many recent works focus on dynamic scenes with the idea of 3DGS (Wu et al., 2024; Luiten et al., 2024; Yang et al., 2024; Duan et al., 2024; Sun et al., 2024). Dynamic3DGS (Luiten et al., 2024) optimize the attributes of existing Gaussians to deal with new frames and perform tracking. 4DGS (Wu et al., 2024) build a multi-resolution voxel planet to compute voxel feature with timesteps. Realtime4DGS (Yang et al., 2024) build a 4D Gaussian structure and condition it to 3D Gaussian with a given timestep. 3DGStream (Sun et al., 2024) focuses on online training and builds a transformation cache for optimization. However, despite being an online method, 3DGStream continuously prunes Gaussians during training, making it impossible to perform 3D point tracking across all time frames. While all these methods benefit from the efficiency of differentiable rasterization, they fail to leverage the internal structural information of the real world and still suffer from notable training time. SC-GS (Huang

et al., 2024) utilizes control points to compress the motion information of Gaussians, but struggles to achieve accurate per-point tracking and highly relies on the distribution of control points.

Our method is mainly inspired by Dynamic3DGS (Luiten et al., 2024) and focuses on the online dynamic scenes (Sun et al., 2024; Li et al., 2022a; Wang et al., 2023; Song et al., 2023), where the method must continually deal with new incoming frames. To make online training much more efficient, we propose a multi-level structure for 3D Gaussians with a new deformation optimization strategy. In addition, our explicit deformation format allows for broad applications like part segmentation.

Compared to Dynamic3DGS (Luiten et al., 2024), our method introduces a key change in deformation modeling: we replace per-Gaussian updates with a coarse-to-fine, multi-layer deformation structure based on clustering. This structural design brings two main advantages. First, by grouping Gaussians with similar motion patterns, the optimization can move larger structures jointly, leading to significantly faster convergence. Second, the multi-layer hierarchy enables refinement at different resolutions: the coarsest layers optimize group-level transformations, while the finest layer retains full per-Gaussian parameter updates. As a result, our method maintains the same level of granularity as Dynamic3DGS (Luiten et al., 2024), without sacrificing the ability to represent details.

**Dynamic Novel-View Synthesis Datasets** for online methods must provide multi-view inputs for each frame. As opposed to offline methods, online methods can only reconstruct one timestep of the scene at a time, with each timestep being initialized using the previous timestep's representation. Therefore, datasets commonly used in offline dynamic synthesis, such as Pumarola et al. (2021) and Park et al. (2021b), cannot be applied in our case. Moreover, our multi-layer, coarse-to-fine design offers a more efficient way to model dynamic Gaussians. It significantly accelerates the convergence during training while preserving the ability to model detailed deformations. Datasets such as Li et al. (2022c) and Broxton et al. (2020), although appearing complex, involve only small-scale movements. As a result, they are not suitable for evaluating our method's capability to model Gaussian dynamics. In the end, we selected accelerated versions of datasets Abou-Chakra et al. (2024) and Luiten et al. (2024) for testing, which meet the aforementioned requirements. For further details, please refer to Sec 4.1.

## 3 Method

**Overview.** In this section, we present the implementation details of our proposed structural cascaded optimization approach. As outlined in the Introduction, our study aims to answer the question: "Can 4D scenes directly benefit from the structural information inherent in vanilla 3DGS?" and to provide insights that could inspire online applications. Offline methods, while effective for high-quality reconstructions, usually do not have straightforward modeling of explicit point-level information. They also lack the potential for online reconstructions. Therefore, we build upon the online method Dynamic3DGS (Luiten et al., 2024) as our codebase and integrate our cascaded optimization approach where each frame's Gaussian outputs are generated solely based on the state of Gaussians from the previous frame and the 2D image inputs from the current frame.

For the complete Dynamic Gaussian Splatting training task, we first perform a static scene reconstruction based on the initial frame observations, following the standard 3D Gaussian Splatting (Kerbl et al., 2023) procedure. Given multi-view observations of a static scene $(I_{0,1}, I_{0,2}, \ldots, I_{0,N})$ and their corresponding camera poses $(C_1, C_2, \ldots, C_N)$, we train a module $\Theta_0$ that represents the parameters of all Gaussians. This module allows us to generate a predicted image $\hat{I}$ for any input camera pose $C$, such that $\hat{I} = \Theta_0(C)$.

Based on this, we can proceed with subsequent online dynamic scene reconstruction. To be more specific, we use $S_0, S_1, \ldots, S_T$ to represent the dynamic scene from time frame 0 to time frame $T$. For each time frame $t$, we have a sequence of images $I_{t,1}, I_{t,2}, \ldots, I_{t,N}$ from the cameras. Our goal is to train a representation $\Theta$ that can fit the scenes $S_0, S_1, \ldots, S_T$. Given an arbitrary camera $C$ at time frame $t$, we can predict the image as $\hat{I} = \Theta_t(C)$.

Throughout this process, we introduce a structural cascaded optimization approach that organizes the Gaussians in a coarse-to-fine manner, significantly reducing the number of training iterations required between consecutive frames. In the following sections, we provide a detailed explanation of each step in this approach.

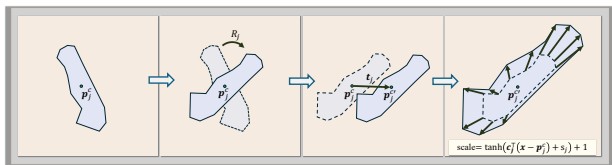

Figure 3: Illustration of the deformation function parameters: Rotation (second), translation (third), and scaling (fourth) applied to a cluster. Initial state (first).

## 3.1 Preliminary

In this section, we outline the key concepts and steps involved in learning the dynamic scene representation $\Theta$ for a sequence of dynamic scenes $S_0, S_1, \ldots, S_T$. For online dynamic scene reconstruction, our method focuses on predicting the deformation between two consecutive frames using the reconstruction results from the previous frame and the current frame's input observations.

Assuming that the Gaussians of frame $t-1$ have been reconstructed, we need to predict the deformation for frame $t$ and obtain the scene representation $\Theta_t$ for it. To be concrete, we want to predict the deformation $D_t$ that satisfies the following equation:

$$\Theta_t = D_t(\Theta_{t-1}). \tag{1}$$

Thus, for any given $t$ and $C$, we have

$$\Theta(t, C) = D_t(D_{t-1}(\cdots D_1(\Theta_0) \cdots))(C). \tag{2}$$

Consider the changes of a single Gaussian $g_i$ in the scene at time frame $t$ during deformation. Recall that its representation is defined as

$$G_{t,i}(\mathbf{X}) = e^{-\frac{1}{2}(\mathbf{X}-\boldsymbol{p}_{t,i})^T \boldsymbol{\Sigma}_{t,i}^{-1}(\mathbf{X}-\boldsymbol{p}_{t,i})}. \tag{3}$$

This is a probability density function of the position $\mathbf{X}$ in which $\boldsymbol{p}_{t,i}$ is the centroid position, and $\boldsymbol{\Sigma}_{t,i}$ is the covariance matrix. In the deformation process $D_t$, we assume that the corresponding deformation function of position $x$ is $\Phi_t$ which satisfies

$$\Phi_t(\boldsymbol{p}_{t-1,i}) = \boldsymbol{p}_{t,i}. \tag{4}$$

According to the derivation in Xie et al. (2024), the eformed centroid position $\boldsymbol{p}_t$ and the covariance matrix $\boldsymbol{\Sigma}_t$ as follows:

$$\begin{aligned} \boldsymbol{p}_{t,i} &= \Phi_t(\boldsymbol{p}_{t-1,i}), \\ \boldsymbol{\Sigma}_{t,i} &= \nabla_{\boldsymbol{p}_{t-1,i}}(\Phi_t)\boldsymbol{\Sigma}_{t-1,i}\nabla_{\boldsymbol{p}_{t-1,i}}(\Phi_t)^T. \end{aligned} \tag{5}$$

This means that if we can learn the deformation function $\Phi_t$ of the scene, we can use Eq. (5) directly to update the parameters of all the Gaussians. Thus, our task is transformed into learning $\Phi_t$, which will be discussed in the following sections.

## 3.2 Single-layer Deformation Function

The deformation function $\Phi_t$ can be a complicated non-linear one for the entire scene, making it hard for us to directly learn it. An intuitive idea is that if we can cluster points that are close in space and make an approximation that all the Gaussians within one cluster follow the same deformation function, then the difficulty of learning the deformation function as a whole will be reduced. Also, with this clustering structure, we can make the learning process more efficient than learning it for each Gaussian independently. Which will be revealed in the experiment results. Furthermore, the deformation function within one cluster can be constructed using deformations such as rotation, translation, and scaling, making it possible to parameterize $\Phi_t$ in an explicit form.

The intuition is that one small chunk of the object is nearly rigid, thus its movement can be represented by a transformation and a rotation around its centroid. Also, to increase the flexibility, we can add a scaling

factor. The deformation function $\Phi_t$ within one cluster $j$ can be represented as

$$\boldsymbol{x}_d = (R_j(\boldsymbol{x} - \boldsymbol{p}_j^c) + \boldsymbol{t}_j) \cdot (\tanh{(\boldsymbol{c}_j^\top (\boldsymbol{x} - \boldsymbol{p}_j^c) + s_j)} + 1), \tag{6}$$

where $\boldsymbol{x}$ is the position of the point, $\boldsymbol{x}_d$ is the corresponding position after deformation, $\boldsymbol{p}_j^c$ is the centroid of the cluster, $R_j$ is the rotation matrix (stored as a quaternion to ensure that it represents a rotation), $\boldsymbol{t}_j$ is the translation vector, and $(\tanh{(\boldsymbol{c}_j^\top (\boldsymbol{x} - \boldsymbol{p}_j^c) + s_j)} + 1)$ as a whole is the scaling factor. The inner term $c_j^\top (x - p_j^c) + s_j$ represents a simple linear mapping from the local coordinate to a scalar, with $c_j \in \mathbb{R}^3$ controlling the direction and sensitivity of the scaling, and $s_j \in \mathbb{R}$ acting as a bias. This design follows the standard form of a linear transformation and introduces no special assumptions, while allowing each cluster to flexibly learn a position-dependent scaling. The tanh design ensures the scaling factor remains in the range $(0, 2)$, preventing potential NaN problems during training. Additionally, the scaling factor is a flexible, trainable linear function of $(\boldsymbol{x} - \boldsymbol{p}_j^c)$, with $\boldsymbol{c}_j$ and $s_j$ as its parameters, allowing for adaptable scaling within a single cluster. Fig. 3 illustrates the specific meaning of each parameter. In summary, to represent the deformation function $\Phi_t$ within cluster $j$, we need to learn trainable parameters $R_j$, $\boldsymbol{t}_j$, $\boldsymbol{c}_j$ and $s_j$.

## 3.3 Cascaded Structural Optimization Strategy

The previously discussed content addresses the deformation formulation problem within a single-layer cluster. However, there is a trade-off: if clusters are too small, they provide limited acceleration, while if they are too large, they group too many Gaussians, reducing the ability to capture detailed deformations. To address this, we use a coarse-to-fine multi-layer cluster structure, starting with K-means clustering based on centroids. Subsequently, by merging neighboring clusters, we acquire a coarser layer of clusters. This process is iteratively repeated until we obtain the coarsest layer of clusters. Specifically, suppose a point $\boldsymbol{p}_{t-1}$ belongs to clusters $j_1, j_2, \ldots, j_K$ at each layer respectively ($K$ is the number of layers), and the deformation function within cluster $j_k$ is $\phi_{k,j_k}$. Then, for the point $\boldsymbol{p}_{t-1}$ at the $t-1$-th frame

$$\boldsymbol{p}_t = \phi_{K,j_K}(\cdots(\phi_{2,j_2}(\phi_{1,j_1}(\boldsymbol{p}_{t-1})))\cdots). \tag{7}$$

In our implementation, $K = 3$. The cascaded optimization framework is shown in Fig. 2.

At the coarsest level, clusters are expected to make broad approximations of the scene's deformation. While this coarse clustering might not always align perfectly with the underlying rigid parts, the purpose is to rapidly bring the Gaussians closer to an optimal solution. Fine-level clusters, operating at higher resolutions, can then start optimization from an improved baseline, requiring fewer iterations to refine the deformation. This hierarchical approach reduces training cost while retaining the ability to express detailed motion.

To further enhance the fine-tuning capability of each Gaussian, we introduce three additional parameters for each Gaussian, which are $\Delta\boldsymbol{p}$, $\Delta R$, and $\Delta\boldsymbol{s}$, corresponding to delta in centroids positions, rotations, and scalings. These delta values are applied to the Gaussians after they have been deformed by the deformation function.

## 3.4 Optimization Process and Training Strategies

**Optimization Pipeline.** Based on the previously discussed deformation process, we present our complete optimization pipeline for learning deformations, which consists of two main stages. In the initialization stage, we first train Gaussians on the static scene using observations from the initial frame, followed by a coarse-to-fine clustering of Gaussian centroids. This clustering generally only needs to be done once but can be updated mid-training if there are significant scene changes. In the training stage, for each subsequent frame, we refine the deformation parameters by combining the current input images with the Gaussians from the previous frame. Through backpropagation of 2D loss, we iteratively update the deformation parameters, using the previous frame's parameters as a starting point to effectively capture gradual changes over time.

**Loss Functions for Deformation.** In addition to the 2D image losses used in most Gaussian Splatting methods, following Luiten et al. (2024), we also use local-rigidity loss, isometry loss, and rotation loss to restrict the movement of Gaussians in large regions of the same color. Furthermore, we add scale loss $L^{\text{scale}}$

| Metrics | Method | FastParticle | | | | | | Panoptic | | | | | |
|---|---|---|---|---|---|---|---|---|---|---|---|---|---|
| | | Robot | Spring | Wheel | Pendulums | Robot-Task | Cloth | Basketball | Boxes | Football | Juggle | Softball | Tennis |
| PSNR↑ | Ours$_{100}$ | **29.46** | **30.28** | **27.95** | **30.6** | 27.67 | **31.68** | **30.25** | **29.46** | **30.47** | **31.12** | **31.02** | **30.21** |
| | Dynamic3DGS$_{100}$ (Luiten et al., 2024) | 21.28 | 23.66 | 24.14 | 24.98 | 23.41 | 21.44 | 29.48 | 29.20 | 30.05 | 30.96 | 30.64 | 29.77 |
| | Dynamic3DGS$_{2000}$ (Luiten et al., 2024) | **30.23** | **30.88** | **28.59** | **31.23** | **29.36** | **32.91** | **30.01** | **29.29** | **30.4** | **31.04** | **30.88** | **30.11** |
| SSIM↑ | Ours$_{100}$ | **0.96** | **0.97** | **0.94** | **0.97** | **0.95** | **0.97** | **0.93** | **0.93** | **0.94** | **0.94** | **0.94** | **0.94** |
| | Dynamic3DGS$_{100}$ (Luiten et al., 2024) | 0.90 | 0.93 | 0.89 | 0.94 | 0.92 | 0.92 | **0.92** | **0.93** | **0.93** | **0.94** | **0.94** | **0.94** |
| | Dynamic3DGS$_{2000}$ (Luiten et al., 2024) | **0.97** | **0.97** | **0.94** | **0.97** | **0.97** | **0.98** | **0.92** | **0.93** | **0.93** | **0.94** | **0.94** | **0.94** |
| LPIPS↓ | Ours$_{100}$ | **0.09** | **0.04** | **0.07** | **0.06** | **0.10** | **0.06** | **0.21** | **0.20** | **0.20** | **0.20** | **0.20** | **0.19** |
| | Dynamic3DGS$_{100}$ (Luiten et al., 2024) | 0.15 | 0.08 | 0.11 | 0.09 | 0.13 | 0.11 | 0.22 | 0.21 | 0.21 | **0.20** | 0.21 | 0.21 |
| | Dynamic3DGS$_{2000}$ (Luiten et al., 2024) | **0.08** | **0.04** | **0.06** | **0.05** | **0.09** | **0.05** | 0.22 | **0.21** | **0.21** | **0.21** | **0.21** | **0.21** |

Table 1: Online methods rendering results on the FastParticle and Panoptic datasets. Values represent mean metrics across all testing views. Top-2 methods are bolded.

| Metrics | Method | FastParticle | | | | | | Panoptic | | | | | |
|---|---|---|---|---|---|---|---|---|---|---|---|---|---|
| | | Robot | Spring | Wheel | Pendulums | Robot-Task | Cloth | Basketball | Boxes | Football | Juggle | Softball | Tennis |
| PSNR↑ | Ours | **29.46** | **30.28** | **27.95** | **30.60** | 27.67 | **31.68** | **30.25** | **29.46** | **30.47** | **31.12** | **31.02** | **30.21** |
| | Dynamic3DGS (Luiten et al., 2024) | 21.28 | 23.66 | 24.14 | 24.98 | 23.41 | 21.44 | 29.48 | 29.2 | 30.05 | 30.96 | 30.64 | 29.77 |
| | RealTime4DGS (Yang et al., 2024) | 25.97 | 22.54 | 23.86 | 26.25 | 24.72 | 22.16 | 25.51 | 27.59 | 26.48 | 27.63 | 26.73 | 27.09 |
| | 4DGS (Wu et al., 2024) | 25.86 | 24.93 | 26.56 | 27.35 | **28.00** | 27.89 | 23.26 | 28.02 | 27.04 | 28.10 | 26.01 | 27.54 |
| | SC-GS(no-pretraining) (Huang et al., 2024) | 15.76 | 17.08 | 16.89 | 17.90 | 16.42 | 14.58 | 19.72 | 21.43 | 20.66 | 20.87 | 21.03 | 21.10 |
| | SC-GS(pretraining) (Huang et al., 2024) | 22.31 | 25.60 | 24.10 | 27.32 | 26.49 | 26.95 | 19.42 | 21.02 | 20.17 | 20.62 | 21.11 | 21.02 |
| SSIM↑ | Ours | **0.96** | **0.97** | **0.94** | **0.97** | 0.95 | **0.97** | **0.93** | **0.93** | **0.94** | **0.94** | **0.94** | **0.94** |
| | Dynamic3DGS (Luiten et al., 2024) | 0.90 | 0.93 | 0.89 | 0.94 | 0.92 | 0.92 | 0.92 | **0.93** | 0.93 | **0.94** | **0.94** | **0.94** |
| | RealTime4DGS (Yang et al., 2024) | 0.93 | 0.91 | 0.89 | 0.93 | 0.93 | 0.91 | 0.89 | 0.92 | 0.91 | 0.92 | 0.91 | 0.92 |
| | 4DGS (Wu et al., 2024) | 0.93 | 0.93 | 0.91 | 0.94 | **0.95** | 0.95 | 0.87 | 0.92 | 0.91 | 0.92 | 0.91 | 0.92 |
| | SC-GS(no-pretraining) (Huang et al., 2024) | 0.75 | 0.78 | 0.80 | 0.66 | 0.76 | 0.73 | 0.69 | 0.70 | 0.69 | 0.71 | 0.72 | 0.70 |
| | SC-GS(pretraining) (Huang et al., 2024) | 0.90 | 0.95 | 0.87 | 0.95 | 0.94 | 0.94 | 0.68 | 0.69 | 0.68 | 0.71 | 0.71 | 0.70 |
| LPIPS↓ | Ours | **0.09** | **0.04** | **0.07** | 0.06 | **0.10** | **0.06** | **0.21** | **0.20** | **0.20** | **0.20** | **0.20** | **0.19** |
| | Dynamic3DGS (Luiten et al., 2024) | 0.15 | 0.08 | 0.11 | 0.09 | 0.13 | 0.11 | 0.22 | 0.21 | 0.21 | **0.20** | 0.21 | 0.21 |
| | RealTime4DGS (Yang et al., 2024) | 0.13 | 0.11 | 0.12 | 0.10 | 0.13 | 0.13 | 0.26 | 0.22 | 0.23 | 0.22 | 0.23 | 0.23 |
| | 4DGS (Wu et al., 2024) | 0.12 | 0.08 | 0.10 | 0.09 | 0.11 | 0.09 | 0.32 | 0.25 | 0.27 | 0.25 | 0.26 | 0.25 |
| | SC-GS(no-pretraining) (Huang et al., 2024) | 0.27 | 0.17 | 0.15 | 0.21 | 0.26 | 0.25 | 0.44 | 0.44 | 0.43 | 0.42 | 0.41 | 0.42 |
| | SC-GS(pretraining) (Huang et al., 2024) | 0.12 | **0.04** | 0.12 | **0.04** | **0.07** | 0.07 | 0.45 | 0.43 | 0.44 | 0.41 | 0.41 | 0.42 |

Table 2: General dynamic methods rendering results on the FastParticle and Panoptic datasets. Values represent mean metrics across all testing views. The best method is bolded.

| Metrics | Method | FastParticle | | | | | | Panoptic | | | | | |
|---|---|---|---|---|---|---|---|---|---|---|---|---|---|
| | | Robot | Spring | Wheel | Pendulums | Robot-Task | Cloth | Basketball | Boxes | Football | Juggle | Softball | Tennis |
| 2D MTE↓ | Ours$_{100}$ | **0.80%** | **0.17%** | **14.88%** | **0.50%** | **0.82%** | **0.24%** | **0.57%** | **0.22%** | **7.64%** | **8.15%** | **0.39%** | **1.72%** |
| | Dynamic3DGS$_{100}$ (Luiten et al., 2024) | 7.84% | 2.26% | 18.53% | 3.51% | 4.42% | 2.33% | 15.85% | 4.95% | 9.29% | 12.42% | 16.43% | 25.19% |

Table 3: 2D tracking results on the FastParticle and Panoptic datasets. Values represent mean metrics across all testing trajectories. The best method is bolded.

to prevent the generation of Gaussians that are excessively large or elongated, helping to reduce artifacts during the deformation process. The explicit form of the scale loss is

$$L^{\text{scale}} = \frac{1}{N} \sum_{i=1}^{N} \max(0, \text{scale}_{i,t} - \text{max\_scale}), \tag{8}$$

where $\text{scale}_{i,t}$ is the scaling vector of Gaussian $i$ at time $t$, and $\text{max\_scale}$ is a hyper-parameter. We use fixed empirical weights of $[0.19, 0.10, 0.19, 0.48, 0.05]$ for rigidity, isometry, rotation, scale, and RGB losses, respectively.

**Training Strategy and Appearance Modeling.** After the first round of the static stage, we fix the opacity and background logit of the Gaussians. For better rendering results, we make the color trainable, allowing it to better adapt to different lighting conditions. Specifically, in terms of appearance modeling, we follow the approach of Dynamic3DGS (Luiten et al., 2024), assigning each Gaussian a trainable 3D RGB vector instead of using spherical harmonics (SH). Additionally, we add a soft RGB loss to constrain the changes in color. Regarding the coarse-to-fine clustering, in our experiments, we use a structure with $K = 3$, where the clusters in the coarser layer are obtained by merging clusters from the finer layer. The finest layer clusters are obtained using KMeans of Gaussian centroid positions. The merging method involves calculating the average centroid position of Gaussians in each cluster, and then performing Agglomerative Clustering based on this. The final numbers of clusters at each layer are $64, 320, 1280$.

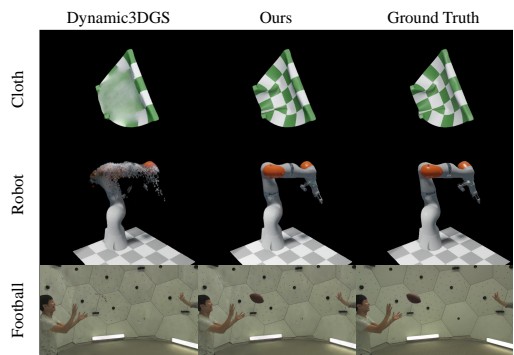

Figure 4: Visual comparison of rendering results on FastParticle after 100 iterations per frame training.

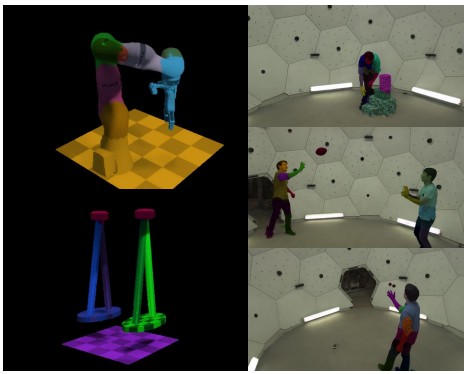

Figure 5: Articulated objects segmentation results.

## 4 Experiments

### 4.1 Dataset Preparation

We conduct our experiments on two datasets: the Panoptic dataset Joo et al. (2015; 2019), which includes six real-world dynamic scenes (Basketball, Boxes, Football, Juggle, Softball, and Tennis), and the synthetic FastParticle dataset (Abou-Chakra et al., 2024), containing six highly dynamic scenes (Robot, Spring, Wheel, Pendulums, Robot-Task, and Cloth). As mentioned in Sec. 2, we deliberately chose these datasets with challenging motion patterns to evaluate our method's ability to quickly converge Gaussians in a very short training period. The large motion between frames in these datasets increases the difficulty of rapid convergence, making them ideal for testing the robustness of our approach. To further amplify this challenge, we accelerated the motion in the FastParticle dataset. Additional details are available in the appendix.

### 4.2 Comparisons

In this section, we demonstrate the effectiveness of our proposed cascaded optimization approach in significantly reducing the number of training iterations required. This is achieved by comparing our method with four state-of-the-art dynamic Gaussian Splatting methods on view-synthesis tasks. These methods include Dynamic3DGS (Luiten et al., 2024), RealTime4DGS (Yang et al., 2024), 4DGS (Wu et al., 2024), and SC-GS (Huang et al., 2024). Notably, Dynamic3DGS (Luiten et al., 2024), which serves as our codebase, follows the same online dynamic scene reconstruction approach as ours, while the other three are offline methods.

**Evaluation Metrics.** We use the PSNR, SSIM, and LPIPS (Wang et al., 2004; Zhang et al., 2018). In the experiments, since training speed is greatly influenced by implementation and hardware, for a fair comparison, it is most reasonable to compare the rendering results at the same iteration number. To eliminate concerns about runtime speed. On our single NVIDIA A40 GPU, training 100 iterations takes 1-3 seconds.

**Comparison with Online Methods.** We conducted experiments comparing our method with Dynamic3DGS (Luiten et al., 2024) on both the FastParticle and Panoptic datasets. The results are shown in Table 1. Since both methods follow the paradigm of first training a static scene and then performing Gaussian Splatting training frame by frame, we fixed the number of training iterations between every two frames to 100 and 2000 for comparison. Here, we provide the same static checkpoints for both methods for fairness. As mentioned earlier, our method significantly reduces the number of iterations required to achieve satisfactory rendering results. From the results in the tables, it can be seen that our method achieves results comparable to Dynamic3DGS (Luiten et al., 2024) at 2000 iterations with only 100 iterations of training between frames, and it far surpasses Dynamic3DGS (Luiten et al., 2024) at 100 iterations. Fig. 4 shows the rendering results of both methods at 100 iterations, qualitatively demonstrating that our method can converge and achieve satisfactory visual results after being trained for only 100 iterations per time frame.

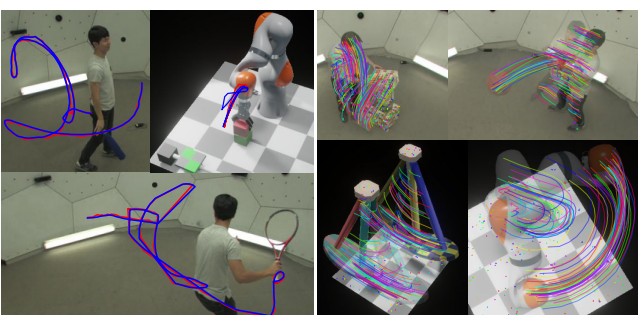 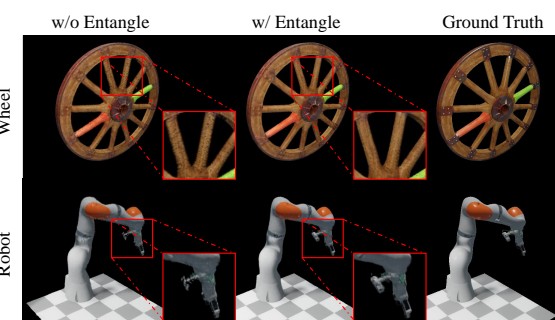

Figure 6: Left: Comparing our tracking result (blue) to the ground truth (red). Right: Visualization of our tracking results.

Figure 7: Ablation study of the entangled covariance matrix.

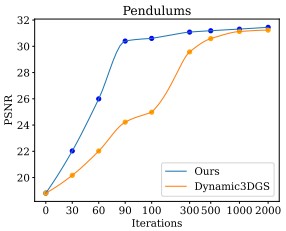 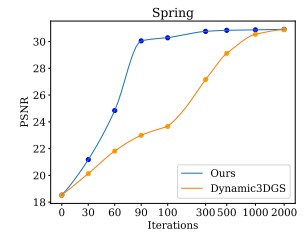 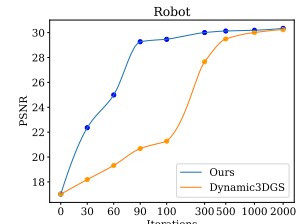

Figure 8: Convergence speed comparison between our method and Dynamic3DGS (Luiten et al., 2024) on the FastParticle dataset. The x-axis shows the number of training iterations per frame, and the y-axis represents the mean PSNR across all testing views.

**Comparison with Online and Offline Methods.** We compared our method with the other four methods on both datasets. For fairness, the two online methods trained 100 iterations per frame, while for the offline methods, we set their total iterations to 100 multiplied by the total number of frames. Similarly, we provided all methods with the same static scene checkpoints for fair comparisons. SC-GS (Huang et al., 2024) is a special case because it involves two stages: the first stage requires 10,000 iterations solely to establish control points, and the second stage begins the actual rendering training. Therefore, we provide two metrics: pretraining refers to the scenario where SC-GS undergoes 10,000 iterations to establish control points before continuing with the same number of iterations as our method, effectively adding 10,000 extra iterations. No-pretraining refers to the case where we skip the additional 10,000 iterations and directly start the rendering training. The results are shown in Table 2. As can be seen from the table, our method achieved the best results across both datasets.

**Point-Tracking.** Additionally, we evaluated our method's point-tracking capability. Due to the challenge of obtaining 3D ground-truth tracking labels, we manually annotated keypoints for all frames from a selected camera view in each scene, using these as ground-truth data. Details of the annotation process are provided in the appendix. For tracking, we projected all Gaussian centroids in each frame onto the camera plane to obtain predicted 2D points. We then selected candidate points within 10 pixels of the ground-truth 2D

| Metrics | Method | Particle | | | | | |
|---|---|---|---|---|---|---|---|
| | | Robot | Spring | Wheel | Pendulums | Robot-Task | Cloth |
| PSNR↑ | Ours, K=3 | **29.46** | **30.28** | **27.95** | **30.60** | **27.67** | **31.68** |
| | Ours, K=1 | 24.56 | 26.16 | 25.12 | 25.94 | 24.73 | 27.77 |
| SSIM↑ | Ours, K=3 | **0.96** | **0.97** | **0.94** | **0.97** | **0.95** | **0.97** |
| | Ours, K=1 | 0.94 | 0.95 | 0.91 | 0.95 | 0.94 | 0.96 |
| LPIPS↓ | Ours, K=3 | **0.09** | **0.04** | **0.07** | **0.06** | **0.10** | **0.06** |
| | Ours, K=1 | 0.12 | 0.07 | 0.10 | 0.08 | 0.12 | 0.07 |

Table 4: Ablation study for the number of cluster layers.

keypoint from the first frame, choosing the one with the highest metric value as the final tracked point. This step was necessary because the candidates corresponded to different depths, and the 2D ground-truth coordinates alone were insufficient for determining which point to track. The candidate with the best metric was considered the 3D-consistently aligned point. We used the 2D Median Trajectory Error (MTE) as the metric, following Dynamic3DGS (Luiten et al., 2024). In Table 3, we report the normalized MTE, which is the pixel error normalized by the image diagonal length, along with visualizations of the tracking results in Fig. 6. We compare our method against Dynamic3DGS (Luiten et al., 2024), selected for its superior rendering performance and as the only baseline aligning with our settings. Our tracking outcomes significantly outperform the baseline across all scenes with the same number of training iterations. Notably, the "Wheel" scene exhibits a high 2D MTE due to the object's strong symmetry, leading to ambiguity in its rotational trajectory (see the appendix for the scene image).

### 4.3 Visualization of Part Segmentation

Next, we demonstrate another application of the learned deformation information: performing segmentation on articulated objects without any semantic knowledge. Many objects in daily life, although not rigid as a whole, are composed of many rigid parts. As humans, we can distinguish these parts by watching a dynamic video and observing their motions. Similarly, our study shows how we can segment different parts of an object based solely on their deformation information, without requiring any semantic knowledge.

After training, we can obtain the centroid positions and rotations of Gaussians at each time frame. We then use KMeans clustering to group Gaussians into different clusters based on this information. Specifically, for a given Gaussian $i$, we use the notations $\boldsymbol{p}_{i,t}$ and $R_{i,t}$ to represent its position and rotation matrix at time frame $t$, respectively. The KMeans feature for each Gaussian is a tensor of shape $[T, 15]$, where $T$ is the total number of time frames. This tensor is the concatenation of $\boldsymbol{p}_{i,t}$, flattened $R_{i,t}$, and $\boldsymbol{p}_{i,0}$ repeated $T$ times. Additionally, we multiply three hyperparameters: $\lambda_{\boldsymbol{p}}$, $\lambda_R$, and $\lambda_{\boldsymbol{p}_0}$ to these three parts before concatenation, respectively, to balance their importance.

The intuition behind the KMeans design is that, (1) Gaussians belong to the same part of the object should be close to each other at all times, and (2) the rotations of Gaussians within the same rigid part should be the same.

As illustrated in Fig. 5, we present our segmentation results on the Panoptic and FastParticle datasets. To enhance visualization, we assign different colors to Gaussians belonging to distinct categories before rendering the final outcomes. Notably, our simple K-means algorithm yields highly intuitive segmentation results, regardless of whether the scene comprises synthetic objects (left) or intricate real-world environments (right). This observation serves as indirect evidence that the deformation information captured by our learned Gaussians closely aligns with the actual motion of objects in dynamic scenes.

### 4.4 Ablation Study

In this section, we conduct ablation studies to analyze the effectiveness of our method. We first analyze the convergence speed of our method and compare it with Dynamic3DGS (Luiten et al., 2024). Then, we study the influence of the number of clustering layers on the rendering results. Finally, we analyze the effect of the entangled covariance matrix on the rendering results.

**Analyse of training iterations** To compare the convergence speed of our method and Dynamic3DGS (Luiten et al., 2024), we trained both methods for different iterations and evaluated their rendering results at these iterations. We trained both methods on the FastParticle dataset. We show the results in Fig. 8, where the x-axis represents the number of training iterations between every two time frames, and the y-axis represents the mean PSNR among all of the testing views. It can be observed that our method converges much faster than Dynamic3DGS (Luiten et al., 2024), consistently outperforming Dynamic3DGS (Luiten et al., 2024) at the same number of iterations. After 2000 iterations, both methods converge at the same point, which also confirms that our method is very close to convergence after training for just 100 iterations.

**Number of Cluster Layers** In our cascaded optimization framework design, we choose the number of layers $K$ to be 3. Here, we conduct experiments to analyze the influence of $K$ on the rendering results, and also to validate the effectiveness of our multi-layer clustering design. We conduct our experiments on the FastParticle dataset, and the results are shown in Table 4. It can be seen that the results of our method with $K = 3$ are much better than those with $K = 1$ across all metrics and scenes, revealing that the coarse-to-fine structure can significantly reduce the number of training iterations, validating the intuition that moving large clusters of Gaussians at once can more quickly find suitable positions, thereby reducing unnecessary adjustments of Gaussian positions. We also test larger values of $K$ (e.g., 4 and 5), and observe no further improvements. In fact, increasing $K$ introduces more parameters to optimize, which may slow convergence. Therefore, $K = 3$ strikes a good balance between efficiency and effectiveness. Detailed results are provided in the appendix.

**Entangled Covariance Matrix** As shown in Eq. (5), in our method, our Gaussians' covariance matrices are not separately learned. Instead, they are coupled with the deformation of centroid positions, following the strategy introduced by PhysGS Xie et al. (2024). This makes it easier for our method to learn the correct rotations and scaling of the Gaussians. In Fig. 7, we show a comparison between learning the covariance matrix separately from positions and our full implementation.

Using the wheel as an example, it should rotate around its own center. As shown in the left, without coupling, although the positions of the Gaussians are learned correctly, the Gaussians themselves do not rotate accordingly with the wheel, resulting in suboptimal final rendering. In our full implementation, as long as the deformation function is learned correctly, the rotations and scalings of the Gaussians are naturally adjusted accordingly, preventing artifacts where the Gaussians are incorrectly oriented.

## 5 Conclusion

In this paper, we presented SCas4D , a cascaded optimization framework for dynamic scene modeling, focusing on efficient tracking and novel-view synthesis. By leveraging hierarchical deformation patterns, our method refines adjustments from part-level to point-level, achieving convergence within 100 iterations while maintaining competitive quality with only 1/20th of the training iterations. We also demonstrated its effectiveness in self-supervised articulated object segmentation. Extensive experiments highlight its strong performance in novel view synthesis and dense point tracking, with potential for further improvement through other 3D Gaussian Splatting variants.

## Acknowledgments

We thank Pavel Tokmakov for valuable discussions. This work was supported in part by NSF Grant 2106825, NIFA Award 2020-67021-32799, the Toyota Research Institute, Amazon, the IBM-Illinois Discovery Accelerator Institute, and Snap Inc. This work used computational resources, including the NCSA Delta and DeltaAI supercomputers through allocations CIS230012 and CIS240370 from the Advanced Cyberinfrastructure Coordination Ecosystem: Services & Support (ACCESS) program, as well as the TACC Frontera supercomputer and Amazon Web Services (AWS) through the National Artificial Intelligence Research Resource (NAIRR) Pilot.

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

# A  Appendix

In this appendix, we offer additional details regarding our FastParticle and Panoptic datasets, which provide the necessary context for our experiments. We delve into our method for articulated objects segmentation, presenting full qualitative results that demonstrate its effectiveness across various scenarios. Additionally, we clarify our rationale for maintaining the same number of iterations in our comparisons and present a comparison under equal wall-clock time, showing that our method still outperforms Dynamic3DGS Luiten et al. (2024). We also include visualizations illustrating the multi-layer clustering structure we employ, as well as the manually annotated tracking labels used for evaluating 2D tracking results. Furthermore, we discuss our approach to learning the deformation, emphasizing the two-phase training strategy. Finally, we reflect on limitations, identifying potential areas for future improvement.

## A.1  FastParticle and Panoptic Datasets

In this section, we introduce the FastParticle and Panoptic datasets used in our experiments in details. The real-world Panoptic dataset includes six scenes: Basketball, Boxes, Football, Juggle, Softball, and Tennis. Each frame in these scenes comes with segmentation provided by the original authors. Following Luiten et al. (2024), we distinguish between foreground and background in these scenes and utilize background loss and floor loss accordingly. Each scene in this dataset contains 150 frames captured by a total of 31 cameras, with 27 cameras used for training and 4 for testing.

The synthetic FastParticle dataset, which we have accelerated, contains six dynamic scenes: Robot, Spring, Wheel, Pendulums, Robot-Task, and Cloth. After acceleration, these scenes respectively have 35, 18, 38, 24, 35, and 35 frames. As illustrated in Fig. 9, we show the dynamic evolution of some scenes, highlighting the significant changes between frames. This dataset includes 40 cameras in total, from which we randomly select 4 as testing cameras and the remaining 36 as training cameras.

For all experiments, we provide the same static checkpoints to all baselines. For the 12 scenes across the two datasets, we train for 20,000 iterations to obtain the checkpoints. Due to the varying complexity of the static scenes, 3,000 iterations are sufficient for most FastParticle scenes.

## A.2  Articulated Objects Segmentation

As mentioned in Sec. 5.1. The intuition behind the KMeans design is that, (1) Gaussians belong to the same part of the object should be close to each other at all time, and (2) the rotations of Gaussians within the same rigid part should be the same. The first one can be trivial, here we provide more explanations about the second point. As shown in Fig. 10, suppose we have a rigid body with its centroid denoted as $C_0$. This rigid body can be considered as a combination of two smaller rigid bodies, with their centroids denoted as $C_1$ and $C_2$, respectively. After rotation, $C_1$ and $C_2$ move to $C_1'$ and $C_2'$. Taking $C_0$ as the origin of the coordinate system, the movement of the rigid body can only be a rotation $R$ around $C_0$, and the two smaller rigid bodies move accordingly. When considering the left smaller rigid body alone, its motion should consist of a translation of its centroid $C_1$ and a rotation $R_1$ around $C_1$. We aim to prove that $R_1 = R$. Therefore, consider a point $P$ on the left rigid body, which moves to point $P'$ after the movement. From the perspective of $C_0$, we have

$$\overrightarrow{C_0P'} = R\,\overrightarrow{C_0P}. \tag{9}$$

Also, from the perspective of $C_1$, we can have

$$\begin{aligned}
\overrightarrow{C_0P'} &= R_1\overrightarrow{C_1P} + \overrightarrow{C_0C_1'} \\
&= R_1\overrightarrow{C_1P} + R\,\overrightarrow{C_0C_1}.
\end{aligned} \tag{10}$$

Therefore, we have

$$R\,\overrightarrow{C_1P} = R_1\,\overrightarrow{C_1P}. \tag{11}$$

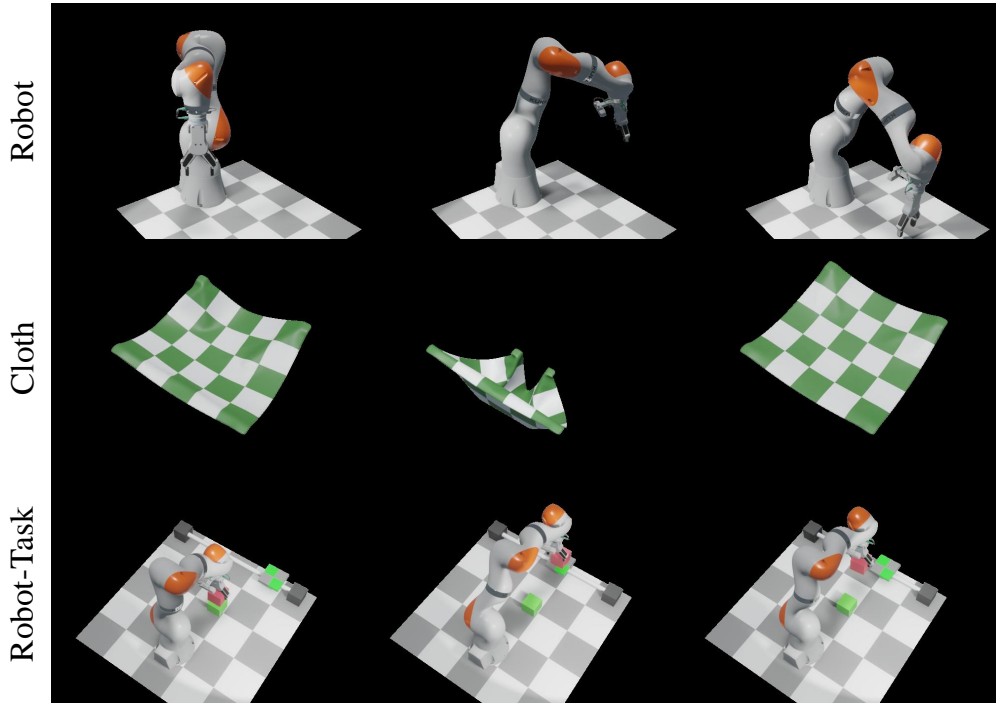

Figure 9: This figure shows the evolution of three scenes from the FastParticle dataset, demonstrating the high dynamic characteristics of the accelerated dataset.

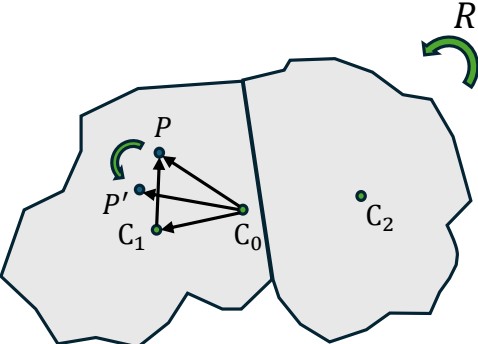

Figure 10: Illustration of a rigid body rotating $R$ around its centroid. When considering the rigid body as composed of two smaller rigid bodies, it can be shown that the rotation of each smaller rigid body around its own centroid is the same with $R$.

Since the choice of $P$ is arbitrary, we can conclude that $R_1 = R$. Similarly, we can prove that the rotation of the smaller rigid body on the right is also $R$. The above demonstrates the case where the rigid body is divided into two parts. This conclusion can be generalized to any case of multiple divisions, meaning that all parts of the same rigid body have the same rotation. Returning to our problem, since the rotation of Gaussians is around their centroids, the Gaussians belonging to the same rigid body should have the same rotation.

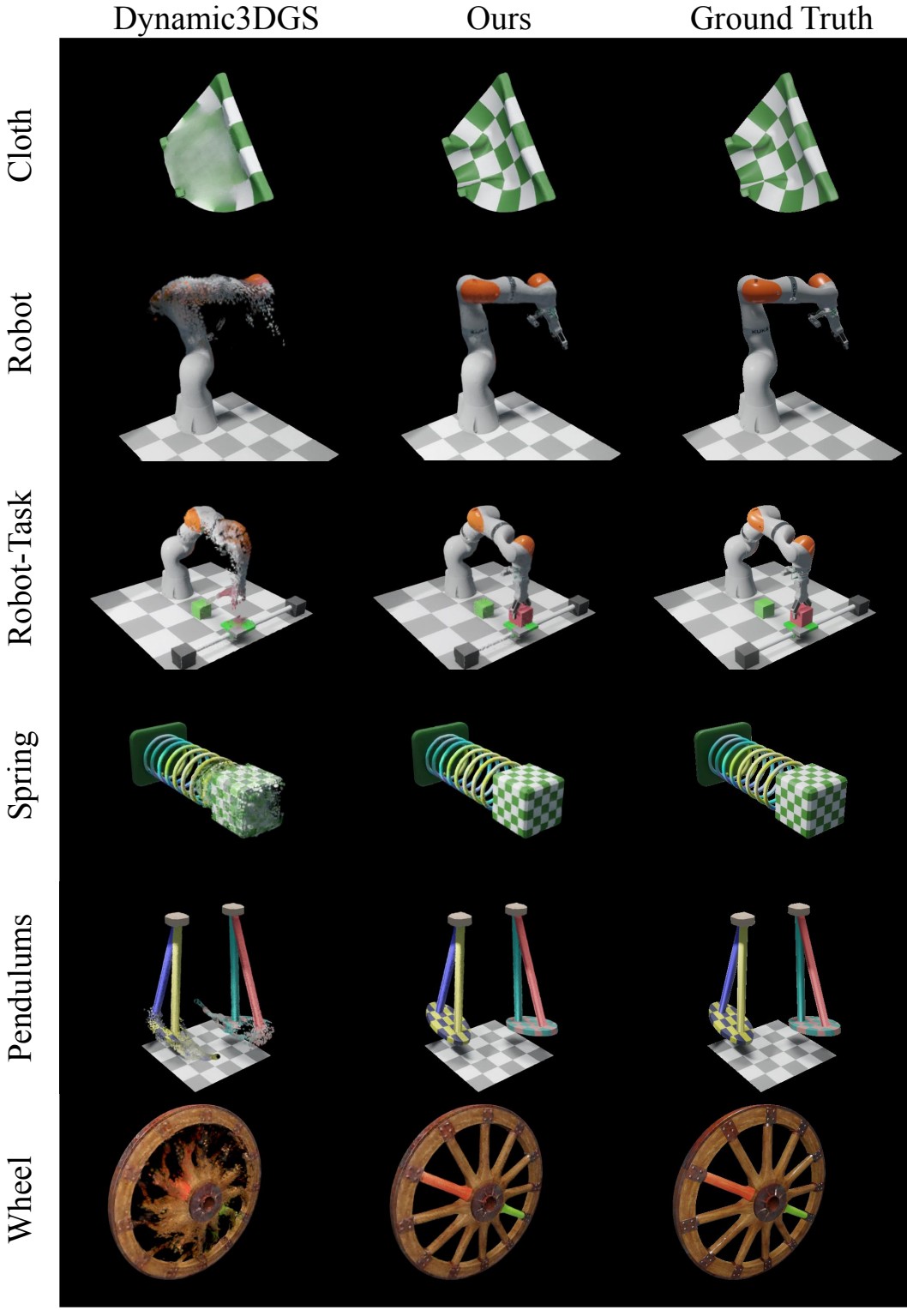

Figure 11: Qualitative results on FastParticle

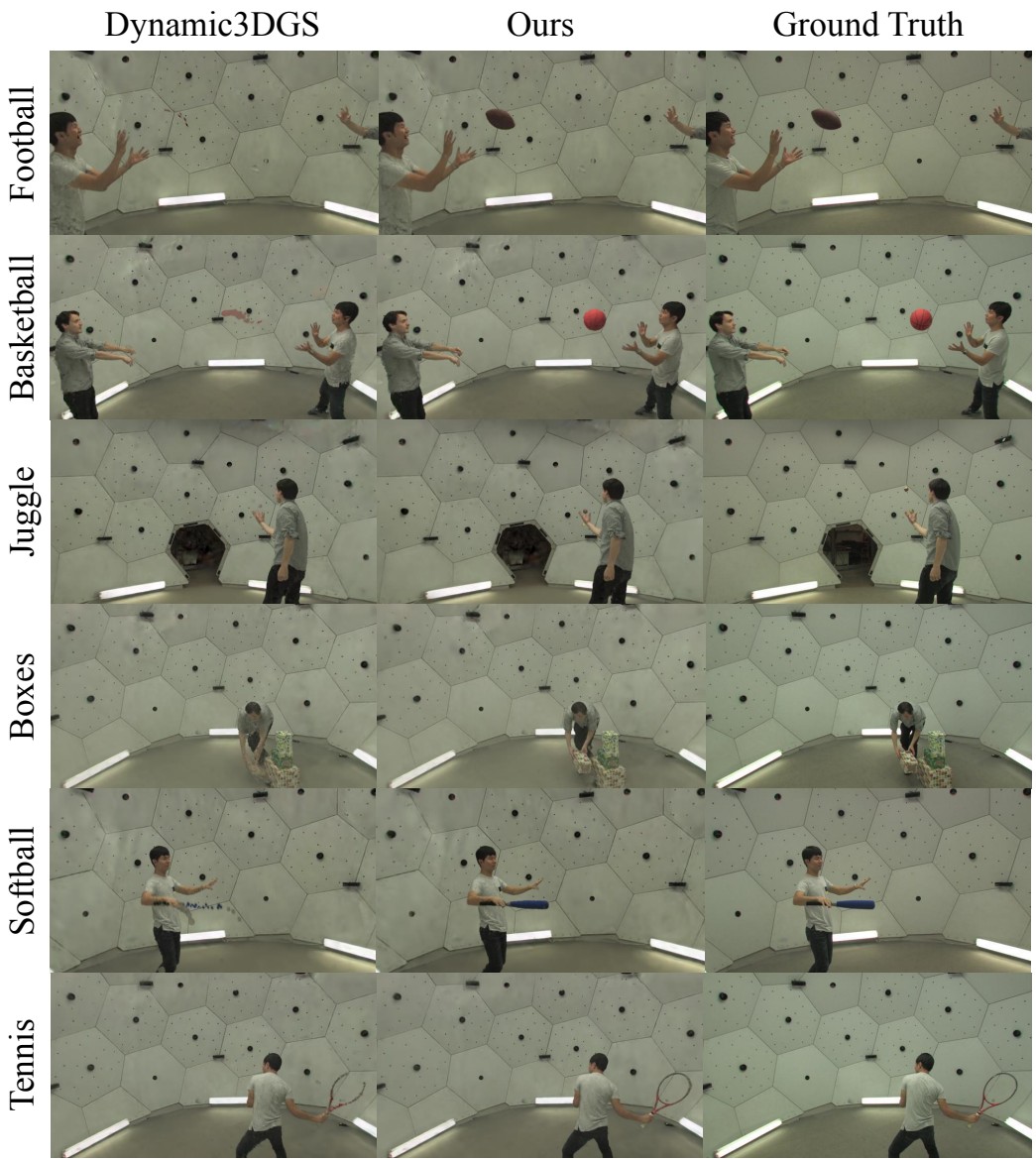

Figure 12: Qualitative results on Panoptic

| Metrics | Method | FastParticle | | | | | |
|---|---|---|---|---|---|---|---|
| | | Robot | Spring | Wheel | Pendulums | Robot-Task | Cloth |
| PSNR↑ | Ours$_{100}$ | **29.46** | **30.28** | **27.95** | **30.60** | **27.67** | **31.68** |
| | Dynamic3DGS$_{300}$ Luiten et al. (2024) | 27.66 | 27.16 | 26.67 | 29.57 | 26.79 | 30.41 |
| SSIM↑ | Ours$_{100}$ | **0.96** | **0.97** | **0.94** | **0.97** | **0.95** | **0.97** |
| | Dynamic3DGS$_{300}$ Luiten et al. (2024) | 0.95 | 0.95 | 0.93 | 0.96 | **0.95** | **0.97** |
| LPIPS↓ | Ours$_{100}$ | **0.09** | **0.04** | **0.07** | **0.06** | **0.10** | **0.06** |
| | Dynamic3DGS$_{300}$ Luiten et al. (2024) | 0.10 | 0.06 | 0.08 | **0.06** | **0.10** | 0.07 |

Table 5: Comparison of our method trained with 100 iterations per time frame against Dynamic3DGS.

| Method | Particle | | | | | |
|---|---|---|---|---|---|---|
| | Robot | Spring | Wheel | Pendulums | Robot-Task | Cloth |
| K=2 | 29.48 | 30.40 | 27.86 | 30.54 | 27.35 | 31.51 |
| K=4 | 29.39 | 30.09 | 27.87 | 30.38 | 27.55 | 31.48 |
| K=5 | 29.17 | 30.00 | 27.79 | 30.36 | 27.60 | 31.43 |

Table 6: More ablation study results for the number of cluster layers. The reported metric is PSNR.

## A.3 Full Qualitative Results

In this section, we provide qualitative results on all 12 scenes from the two datasets. As shown in Fig. 11 and Fig. 12, both our method and Luiten et al. (2024) are trained 100 iterations between two consecutive frames.

## A.4 Same Wall-clock Time Comparisons

In our experiments, we use the same number of iterations across different methods for consistency. While wall-clock time may vary depending on the specific implementation (e.g., whether CUDA acceleration is employed), the number of iterations reflects the convergence speed of the algorithms. A lower number of iterations indicates faster convergence, showing that the optimization problem is easier to solve. This practice is commonly used in the evaluation of online methods, as demonstrated in the Dynamic3DGS Luiten et al. (2024) comparison (see Table 1 in their paper), where different methods are also compared using the same number of iterations.

Even when comparing with equivalent wall time, our method remains superior. To further illustrate this, we provide a comparison of our method trained for 100 iterations per frame versus Dynamic3DGS Luiten et al. (2024) trained for 300 iterations per frame on the FastParticle dataset. The results show that our method has an average training speed per iteration approximately twice as fast as Dynamic3DGS Luiten et al. (2024). As seen in Table 5, despite the difference in iteration count, our method still outperforms Dynamic3DGS Luiten et al. (2024) in terms of both efficiency and final performance.

## A.5 Ablation Study on Number of Cluster Layers

We test layer numbers $K$ from 1 to 5 on the FastParticle dataset. As shown in Table 6, $K = 3$ performs best. We find that increasing the number of cluster layers from 3 does not bring additional performance gains. On the contrary, it introduces more parameters to optimize, which may increase the required number of iterations and lead to diminishing returns. Therefore, we conclude that $K = 3$ offers the best trade-off between efficiency and performance in our framework.

## A.6 Illustration of the Multi-Layer Structure

In Fig. 13, we show the coarse-to-fine multi-layer clustering structures for two objects in the FastParticle dataset. Different colors in the figure represent different clusters, and for clarification, the same color in different layers does not indicate any correlation between the clusters.

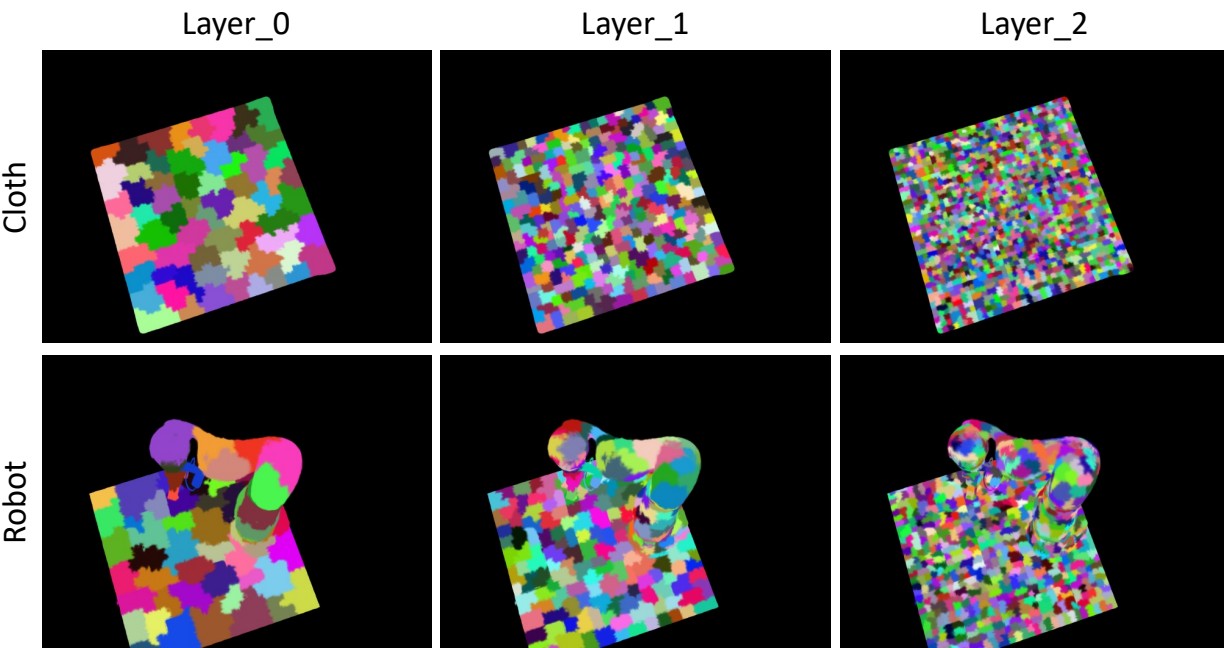

Figure 13: Coarse-to-fine multi-layer clustering structures for two objects in the FastParticle dataset.

### A.7 Tracking Labels

Here, as shown in Fig. 14, we present all manually annotated 2D tracking ground truths. Since the human eye can only track points with distinct features across multiple frames, we only selected such points for annotation.

### A.8 Learning the Deformation

Algorithm 1 summarizes our training process. Initially, we train our Gaussians on the static scene using observations from the first frame. Subsequently, we perform multilevel coarse-to-fine clustering for the centroids of the Gaussians. For each input in every time frame, we use an optimization approach to backpropagate loss and subsequently update our deformation functions.

---

**Algorithm 1:** Deformation-based Dynamic Scene Reconstruction Algorithm

---

**Input:** Images from all frames
$\Theta_{\text{prev}} \leftarrow$ Initialization stage (Static Gaussian Splatting);
Do Clustering;
**for** $t$ **$in$** $time\_frames$ **do**
    Initialize the Deformation $D$;
    **for** $iter$ **$in$** $max\_iters$ **do**
        $\Theta_{\text{curr}} \leftarrow D(\Theta_{\text{prev}})$;
        Images $\leftarrow$ Render($\Theta_{\text{curr}}$);
        loss $\leftarrow$ Loss(gt\_Images, Images);
        Backpropagate(loss);
    **end**
**end**

---

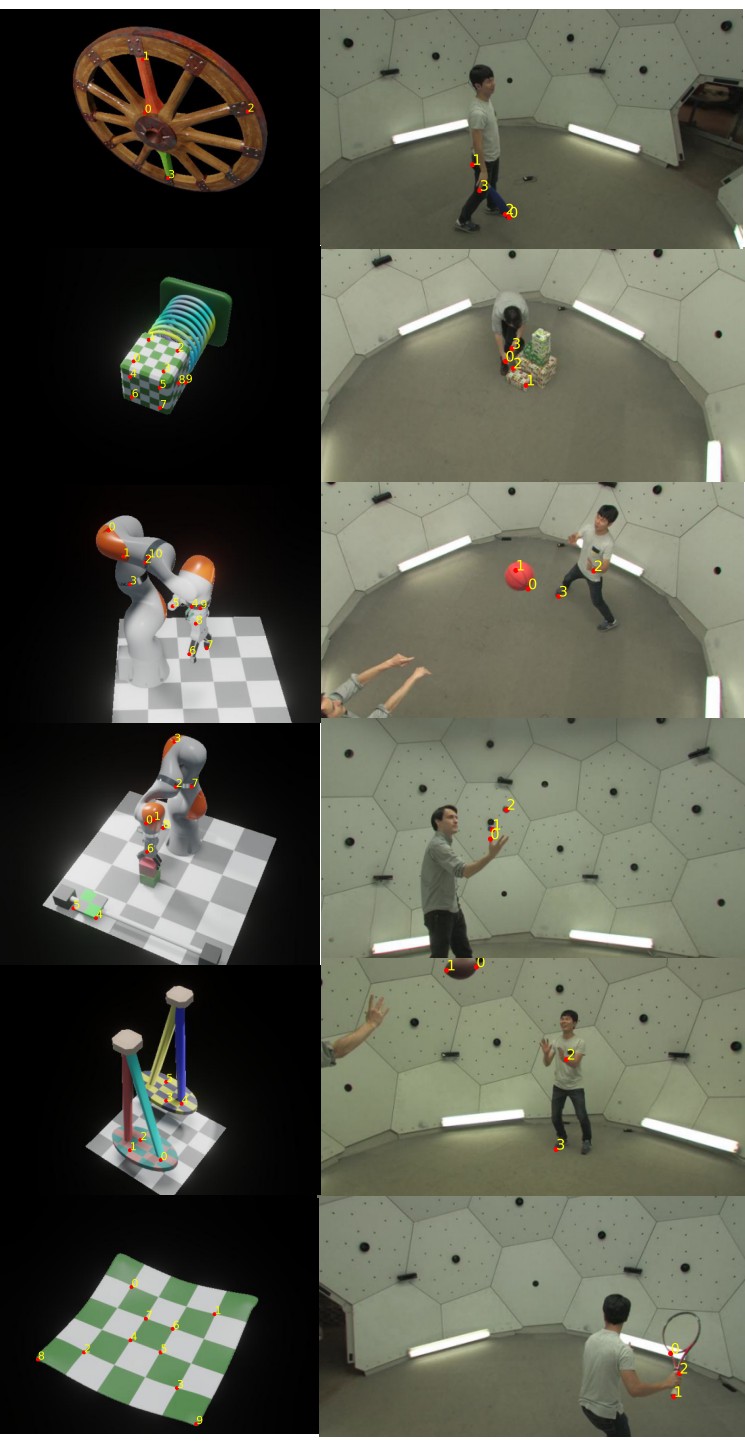

Figure 14: Illustration of our manually annotated tracking ground truths.

| Method | Particle | | | | | |
|---|---|---|---|---|---|---|
| | Robot | Spring | Wheel | Pendulums | Robot-Task | Cloth |
| N=16 | 29.76 | 30.43 | 27.97 | 30.84 | 27.69 | 31.56 |
| N=32 | 29.63 | 30.43 | 27.91 | 30.77 | 27.87 | 31.62 |
| N=48 | 29.49 | 30.36 | 28.00 | 30.70 | 27.62 | 31.61 |

Table 7: PSNR results when varying the number of clusters in the first layer.

For potential negative impacts, since LayeredGS can learn deformation information and be used for creating new motions or inserting objects, such applications can be used for fake news to convince people by multi-view renderings. More censorship needs to be established in such cases.

### A.9 Robustness to the number of clusters in the first layer

To further examine the robustness of our method to the initial clustering configuration, we conducted an additional experiment by varying the number of clusters in the first layer. The default number of clusters used in the main experiments is $N = 64$. Reducing the number of clusters leads to coarser groupings, potentially introducing more clustering errors at the coarse level. Despite this, our model achieves comparable performance across different settings, as summarized in Table 7. These results indicate that our structural cascaded optimization remains robust even when the initial clustering is coarser, thanks to the refinement provided by the multi-layer hierarchy and per-Gaussian updates at the finest level.

### A.10 Effect of the Max Scale Constraint

To further examine the role of the max scale constraint in our method, we conducted an experiment by varying the value of `max_scale`. The scale loss is designed to prevent Gaussians from becoming excessively large during deformation, which may otherwise introduce rendering artifacts. We found that setting a reasonable threshold (in our main experiments, `max_scale` = 0.02) is sufficient to maintain rendering quality.

Our experiment shows that disabling this constraint (i.e., setting `max_scale` to a very large value) leads to significant PSNR degradation, as summarized in Table 8. These results demonstrate the importance of the max scale constraint in preserving stability during optimization.

| Method | Particle | | | | | |
|---|---|---|---|---|---|---|
| | Robot | Spring | Wheel | Pendulums | Robot-Task | Cloth |
| max_scale=0.02 | 29.46 | 30.28 | 27.95 | 30.60 | 27.67 | 31.68 |
| max_scale=2.0 | 15.86 | 19.86 | 22.42 | 18.86 | 18.15 | 21.53 |

Table 8: PSNR results under different values of the max scale constraint.

### A.11 Additional Experiment on Per-Gaussian Deformation for Segmentation

We provide an additional experiment to examine the difference between our method and Dynamic3DGS Luiten et al. (2024) for segmentation purposes. Specifically, we trained Dynamic3DGS Luiten et al. (2024) for 2000 iterations and extracted per-Gaussian deformation (without any layered structure) to perform segmentation, similar to our approach. While this method produces visually reasonable motion, we found that the per-Gaussian rotations are significantly noisier, leading to worse segmentation quality.

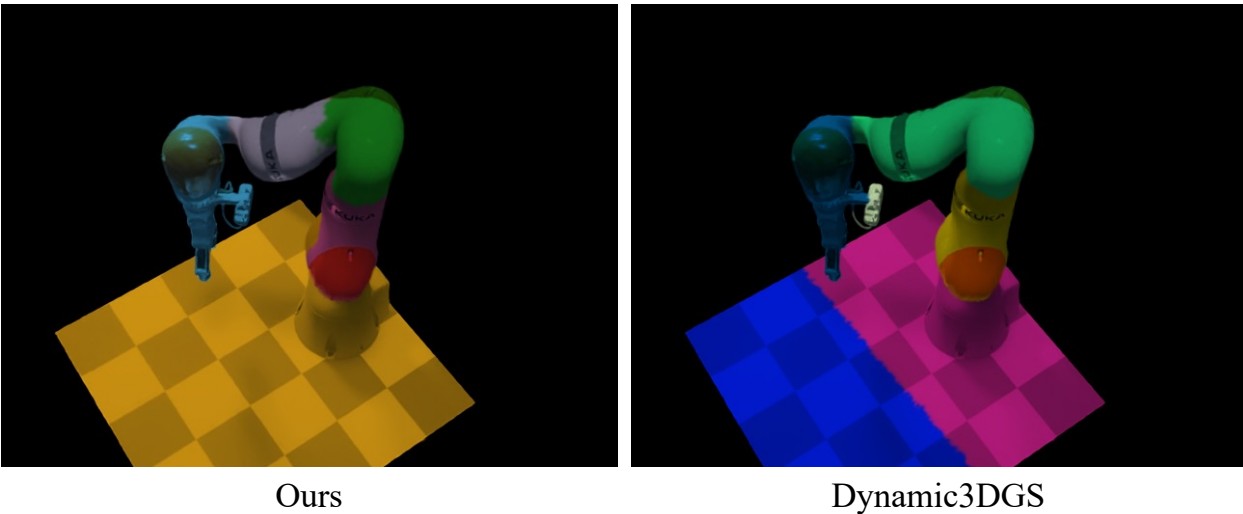

Ours                      Dynamic3DGS

Figure 15: Visual comparison of segmentation results using per-Gaussian deformation (Dynamic3DGS Luiten et al. (2024)) and our multi-layer deformation structure.

In contrast, our multi-layer structure encourages Gaussians to move coherently with their parent clusters, resulting in more stable and interpretable rotations. This structural regularization improves segmentation accuracy by reducing noise in the estimated motion. Please refer to Figure 15 in the updated appendix for visual comparisons.

### A.12 Limitations

While our method significantly reduces training iterations to 100 per frame, achieving real-time training and rendering remains a challenge. Additionally, the learned deformation information is not fully utilized, and the presented articulated object segmentation results are not well refined. Future work will focus on addressing these limitations by exploring real-time training approaches, refining deformation utilization techniques, and developing more sophisticated segmentation methods.

