# OpenReview forum: "SCas4D: Structural Cascaded Optimization for Boosting Persistent 4D Novel View Synthesis"
_TMLR — Accepted by TMLR_

### Review · Reviewer_QwXg · 2025-03-28

**Summary Of Contributions:**

This paper aims to balance the efficiency and quality for 4D view rendering. The core contribution is the insight that every point in a rigid body should be the same. This powerful inductive bias can greatly enhance the rendering efficiency while preseving the quality.

Following this insight, this paper proposed a coarse-to-fine rendering method. Specifically, the coarse level rendering optimizes the whole rigid object using the same transformation. Although this can be very coarse and suboptimal, this procedure can efficiently move all optimzed gaussians to near optimal places. After this, fine level clusters opterate in higher resolutions to refine the overall rendering quality.

The evaluations in terms of rendering quality and efficiency demonstrate the effectiveness of the proposed method, showing a good balance among these two important criterias.

**Audience:**

Yes

**Claims And Evidence:**

Yes

**Requested Changes:**

- Please discuss more about the differences to Dynamic3DGS.

- Why do the K=1 results in Table 4 surpass Dynamic3DGS in Table 1? K=1 can lost most fine-grained information and the quality may degenerate severely.

- What will happen if K is further increased? Will it lead to a performance degeneration? Maybe more ablation studies can be performed.

**Strengths And Weaknesses:**

## Strength

- This paper offers a novel and strong inductive bias that the movements of all points in a rigid body can be describled as a single-layer deformation function.

- This paper proposes a simple yet effective approach to leverage the insights to greatly enhance the training efficiency while maintaining quality.

- The empirical evaluations significatly outperform baselines, maintaining onpar performance while reducing the optimizing steps for online rendering.

- This paper is well-written, clearly presenting its motivation, contribution, and significance.

## Weaknesses

I'am not that falimilar with this area, even not falimilar with the rendering community. So, maybe I cannot offer very valuable comments regarding the weaknesses.

- In my view, one potential weakness is the introduced inductive bias only applys for rigid body, not for deformable objects. So, it would be interesting to see the following work to decrease the deformable objects rendering cost.

- I notice that this paper builds on top of Dynamic3DGS, but dose not introduce the details about it. It would be good if the authors could discuss more about the differences compared to Dynamic3DGS.

---

> ### Author Response · Authors · 2025-05-04
> **Thanks for your review**
>
> We thank the reviewer for the constructive feedback. Below, we provide detailed responses to each concern:
>
> 1. Clarification on differences from Dynamic3DGS
>
> We thank the reviewer for the suggestion. While our method builds upon the Dynamic3DGS pipeline and Gaussian formulation, it introduces a key change in the deformation modeling: we replace per-Gaussian updates with a coarse-to-fine, multi-layer deformation structure based on clustering.
>
> This structural design brings two main advantages. First, by grouping Gaussians with similar motion patterns, the optimization can move larger structures jointly, leading to significantly faster convergence. Second, the multi-layer hierarchy enables refinement at different resolutions: the coarsest layers optimize group-level transformations, while the finest layer retains full per-Gaussian parameter updates. As a result, our method maintains the same level of granularity as Dynamic3DGS, without sacrificing the ability to represent details.
>
> We also propose an additional application—articulated object segmentation—enabled by the learned deformation. We have included the differences in the revised paper. (Section 2)
>
> 2. Clarification on K=1 results
>
> We thank the reviewer for pointing this out. In our method, regardless of the value of K, we always retain per-Gaussian optimization at the finest level. That is, even when K=1, each Gaussian still has its own trainable local refinement parameters (for position, rotation, and scale). Therefore, when K=0 (no coarse structure), our framework reduces to Dynamic3DGS.
>
> Therefore, although K=1 is not the optimal setting, it still benefits from both coarse grouping (accelerating convergence) and fine-grained per-Gaussian corrections. This allows K=1 to achieve competitive performance compared to Dynamic3DGS under limited training iterations. As shown in our ablations (Table 6), introducing more layers (e.g., K=3) further improves performance by providing better coarse-to-fine modeling.
>
> 3. Impact of increasing K
>
> We thank the reviewer for the question. As shown in our ablation study (Appendix A.5, Table 6), increasing K from 1 to 3 improves performance, as more layers allow for better coarse-to-fine deformation modeling. However, further increasing K beyond 3 leads to slight degradation. This is because a larger number of layers introduces more parameters to optimize, while the number of training iterations per frame is kept fixed, making it difficult to fully optimize all layers. Therefore, we choose K=3 as a balance point.
>
> 4. Additional Comment: Inductive bias and deformable objects
>
> We appreciate the reviewer’s comment. We would like to clarify that although our method introduces a structural bias inspired by rigid body motion at the coarse levels, the overall framework still supports non-rigid deformations at finer levels. Each Gaussian retains its own trainable local transformations (position, rotation, scaling), allowing flexible modeling of deformable motion.
>
> We also note that our experimental results already include examples involving significant non-rigid deformations. Specifically, in Figure 4, the Cloth scene and the Football scene (with deforming clothing) demonstrate that our method can effectively handle non-rigid object dynamics.
>
> We have uploaded a revised main paper and appendix (included in the supplementary materials). For clarity, we have temporarily marked all revised texts in purple. The color changing will be removed and reverted to black text in the final version.

---

### Review · Reviewer_WWuz · 2025-04-08

**Summary Of Contributions:**

This paper proposes a multi-layer clustering-based structured optimization for representing dynamic scenes with 3D Gaussians. The authors claim the need to leverage the assumption of locally rigid transformation for dynamic scenes. Given a 3DGS reconstruction for the initial static state (scene) and the subsequent frames' multi-view observations, the proposed system optimizes learnable 3DGS deformation parameters, including rotation, translation, and position-aware scale parameters for 3D Gaussians. Instead of modeling a dynamic scene into a single coarse rigid component, the proposed system models dynamics in a coarse-to-fine manner, where they perform K-means clustering based on the centroids and merge them into bigger clusters (3 levels in practice).

In experiments, the proposed method achieves competitive reconstruction quality compared to the baseline method (Dynamic3DGS), even with 1/20th of the training iterations. Furthermore, the authors built reasonable experimental setups to compare with some offline dynamic reconstruction methods and achieved superior reconstruction quality. Interestingly, when K-means clustering for 3DGS parameters is performed on the final dynamic scene, it can natively segment articulable parts of the dynamic objects.

**Audience:**

Yes

**Claims And Evidence:**

Yes

**Requested Changes:**

For the revised manuscript, this reviewer expects changes as follows (per the weakness comments):
- **Discussions about the effect of the proposed structured optimization strategy on the articulated object segmentation.** Is the structured optimization a crucial requirement for achieving such articulated object segmentation? If not, how does it compare to the articulated object segmentation results that one could get from the baseline method (Dynamic3DGS)?
- **Providing more details about the convergence speed comparison.** How does the PSNR plot look in early iterations or early time frames?
- **Clarification about entangled covariance matrix experiment.** Provide more explicit examples to show the difference and clarify if the covariance matrix entanglement is a proposed contribution or not.
- **Detailed explanation for Eq. (6).** Please provide more details and definitions.
- **Fixing wrong citation.** Please add the right citation for the dataset.

**Strengths And Weaknesses:**

**[Strengths]**
- **The paper is well organized, and the writing is clear.** The main contents (figures, tables, writing) were clear and self-inclusive. This reviewer could find almost all the necessary information from the manuscript. Also, most visual presentations support the claims of the paper.
- **Benefits of structural optimization.** The proposed structural cascaded optimization for dynamic 3DGS optimization is technically sound, and the numbers and figures clearly support it. Also, it is interesting to see that this framework can inherently help intuitive articulated object segmentation.
- **Well-designed experiments.** The authors clearly stated the differences between the online and offline methods for the task. Accordingly, the experiments are clearly designed to make fair comparisons, especially between the proposed method and offline methods. Specifically, the comparison with SC-GS (no-pretraining), SC-GS (pretraining) is clearly explained and makes sense.
- **Included details** This reviewer could find almost all the details and rationale for the design choices from the manuscript. As the proposed method is built on top of the previous work Dynamic3DGS, which is already open-sourced, this reviewer thinks that the proposed technique can be easily followed up by the readers.


**[Weaknesses]**

This reviewer could not find significant weaknesses or errors in the paper. Below are some questions and minor concerns:

- **Question about articulated segmentation.** If the articulated object segmentation is based on the K-means clustering of the final dynamic 3DGS parameters, this reviewer thinks it could be also applied to the competing methods. How will it work with other methods, e.g., Dynamic3DGS? Is this task possible because of the proposed structured optimization technique? It would be interesting to report the differences with and without the proposed structured optimization.
- **Question about Fig. 8: Convergence speed comparison.** While the initial scene states are identical for both Dynamic3DGS and Ours, the PSNR plot shows a significant gap between both methods. This reviewer is aware that the plot shows from iter 100 to 2000. However, this reviewer thinks that showing the plot from iter. 0 would be more compelling as it can highlight the benefit of the proposed method. How do the PSNR plots for iter 0-100 look? Plus, on which temporal frame have the PSNRs computed? This reviewer postulates that the gap should be small for the early frames, as both methods start from the same initial scene.
- **Question about entangled covariance matrix ablation study.** This reviewer cannot see significant visual discrepancies in Fig. 7 w/ and w/o entangle. Also, it seems like the technique to entangle the Gaussian's covariance matrix is from PhysGaussian[1], not this paper's contribution. Could the authors clarify this matter?
- **Detailed explanation for Eq. (6).** As far as this reviewer understood, the scale part, i.e., $(\texttt{tanh}(\mathbf{c}^{\top}\_{j}(\mathbf{x}-\mathbf{p}\_{j}^{c})+s_{j}) + 1)$, of the Eq. (6) allows non-linear deformation with a Gaussian cluster. However, the detailed explanation about this term is missing. Specifically, the definitions for the trainable parameters $\mathbf{c}\_{j}$ and $s\_{j}$ are missing.
- **Wrong citation.** The authors missed to cite the original reference for the Panoptic Studio dataset. It should be citing [2] and [3].


[1] Xie et al., "PhysGaussian: Physics-Integrated 3D Gaussians for Generative Dynamics," CVPR 2024.

[2] Joo et al., "Panoptic Studio: A Massively Multiview System for Social Motion Capture," ICCV 2015.

[3] Joo et al., "Panoptic Studio: A Massively Multiview System for Social Interaction Capture," TPAMI 2017.

---

> ### Author Response · Authors · 2025-05-04
> **Thanks for your review**
>
> We thank the reviewer for the thoughtful and detailed feedback. Below, we provide responses to each point raised:
>
> 1. Articulated segmentation
>
> We thank the reviewer for this question. We provide an additional experiment, please refer to Figure 15 in the updated appendix: We trained Dynamic3DGS for 2000 iterations and extracted per-Gaussian deformation (without any layered structure) to perform segmentation, similar to our approach. While this method does produce visually reasonable motion, we found that the per-Gaussian rotations are significantly noisier, leading to worse segmentation quality. In contrast, our multi-layer structure encourages Gaussians to move coherently with their parent clusters, resulting in more stable and interpretable rotations. This structural regularization helps improve segmentation accuracy.
>
> In each result, different colors indicate different clusters. As shown in Figure 15, our method correctly segments the articulated structure into meaningful parts, while Dynamic3DGS mistakenly assigns the static blue floor and the moving end of the robot arm into the same cluster, and also incorrectly merges a joint of the arm with part of the arm segment (green cluster).
>
> 2. Convergence speed comparison
>
> We appreciate the reviewer’s suggestion. As shown in our new experiment (please refer to Figure 8 in the revised paper), PSNR improves steadily over the first 100 iterations, though our method has not yet converged at that stage.
>
> We chose not to include the iteration-0 point in the main plot because it corresponds to no deformation applied—i.e., directly reusing the static Gaussians from frame 0 to render all subsequent frames. In this case, our method produces exactly the same results as Dynamic3DGS, since both use the same set of static Gaussians. However, this comparison is not informative, as it reflects no motion modeling. Our original plot focuses on iterations where deformation optimization is actively progressing.
>
> 3. Entangled covariance matrix ablation study
>
> We thank the reviewer for raising this point. In Figure 7, we aim to show that enabling entangled covariance matrices ensures that Gaussians undergo not only correct translational motion but also consistent updates in their rotations and scaling.
>
> In the "Wheel" example, although the overall rendering without entanglement appears acceptable, zoomed-in views reveal that the Gaussians' orientations are incorrect: they are perpendicular to the radial direction instead of aligned along it. This indicates that the covariance matrices were not properly updated during deformation.
>
> Similarly, in the "Robot" scene, the end effector becomes blurred without covariance entanglement, highlighting the importance of consistent rotation and scaling updates. Regarding the relation to PhysGS, we acknowledge that our current description may cause confusion. This is not our contribution. The technique of entangling covariance matrices with deformation was first introduced in PhysGS. In our work, we incorporate this design to emphasize its importance for achieving physically consistent deformation. Please refer to section 4.4 in our revised paper, where we clearly state that this is PhysGS’s contribution.
>
> 4. Clarifications on Eq. (6)
>
> We thank the reviewer for the question regarding Eq. (6). We provide a detailed clarification below:
> The term inside Eq. (6), $c_j^T (x - p_j^c) + s_j$, is a simple linear mapping from the local coordinate $(x - p_j^c)$ to a scalar. The trainable parameters are defined as:
> - $c_j$ in $R^3$: a vector controlling the direction and strength of the scaling variation.
> - $s_j$ in $R$: a scalar bias term adjusting the baseline.
>
> This design follows the standard form of linear transformation, with no special assumptions. Our goal is to enable a trainable mapping from position to scaling magnitude in a flexible way. Applying a tanh nonlinearity after the linear mapping serves two purposes: (1) it introduces smooth non-linearity, and (2) it restricts the resulting scaling factor to a bounded range (0,2), ensuring numerical stability during optimization.
>
> We have included these clarifications around Eq. (6) in the revised manuscript.
>
> 5. Incorrect citation
>
> We thank the reviewer for pointing out the incorrect citation. We have corrected this in our revised paper.
>
> We have uploaded a revised main paper and appendix (included in the supplementary materials). For clarity, we have temporarily marked all revised texts in purple. The color changing will be removed and reverted to black text in the final version.

---

### Review · Reviewer_JCjr · 2025-04-20

**Summary Of Contributions:**

This paper introduces SCas4D, a structural cascaded optimization framework that enhances 4D novel-view synthesis and dynamic scene modeling efficiency. SCas4D leverages internal structural information in 3D Gaussian Splatting (3DGS) to significantly speed up convergence rates. It employs a cascaded, multi-level optimization strategy that hierarchically captures deformation from coarse-level to fine-grained adjustments, and introduces a self-supervised articulated object segmentation method. Experimental results demonstrate SCas4D's ability to drastically reduce training iterations while maintaining high rendering quality and superior performance.

**Audience:**

Yes

**Broader Impact Concerns:**

I don't have broader impact concerns.

**Claims And Evidence:**

Yes

**Requested Changes:**

More analysis of clustering on the effect of the performance and other hyperparameters.

**Strengths And Weaknesses:**

Strengths:

1:**The optimization framework is efficient.** The proposed cascaded optimization strategy significantly reduces the training iterations required per frame (only 100 iterations) compared to existing methods (Dynamic3DGS, which requires around 2000 iterations), achieving comparable or superior rendering quality.

2: **Utilization of Structural Information**. SCas4D leverages inherent structural patterns within dynamic scenes by hierarchically organizing 3D Gaussians into clusters. This coarse-to-fine approach effectively captures both large-scale and fine-grained deformations, resulting in improved tracking and segmentation accuracy, particularly valuable for articulated object segmentation tasks.

3: **The proposed method can be applied to many tasks**. Beyond novel-view synthesis, the method demonstrates versatility through applications in dense point tracking and self-supervised articulated object segmentation.

Weaknesses:

1: **Dependence on Initial Clustering.** The method heavily relies on an initial K-means clustering to structure the cascaded optimization. Poor initial clustering can significantly impact subsequent deformation refinement, potentially limiting the method's performance, especially in scenarios involving complex or ambiguous motions.

2: **More in-depth analysis of hyperparameters**. The approach involves multiple hyperparameters, such as the number of clustering layers (the authors only test K=1 and K=3), the maximum scale constraints, and balancing weights in loss functions.

3: **Potential extension of current pipeline**. Despite achieving impressive efficiency gains, the proposed method still cannot achieve full real-time training and rendering, which remains an essential barrier for broader applications in interactive or live systems.

Overall, I think this paper is solid with comprehensive experiments.

---

> ### Author Response · Authors · 2025-05-04
> **Thanks for your review**
>
> We thank the reviewer for the constructive comments. Below, we provide detailed responses to each concern:
>
> 1. Dependence on initial clustering
>
> We thank the reviewer for raising the question regarding our method’s potential sensitivity to the initial clustering configuration. We would like to clarify that the design of our method reduces such sensitivity through two mechanisms:
> - First, as described in Section 3.3 of the main paper, we adopt a multi-layer hierarchical clustering structure, where coarser clusters provide a rough deformation initialization and finer layers progressively refine these deformations.
> - Second, each Gaussian is associated with trainable local adjustments in position, rotation, and scale. This allows the model to recover from potential inaccuracies introduced by imperfect coarse-level clustering.
>
>
> To further examine this point, we conducted an additional experiment in which we varied the number of clusters at the first layer. Reducing this number results in coarser clusters, potentially increasing the chance of grouping errors. Nonetheless, the model achieves comparable performance across different settings, as summarized in the table below:
>
> | Method  \ Scene       | Robot | Spring | Wheel | Pendulums | Robot-Task | Cloth |
> |----------------|-------|--------|-------|-----------|------------|-------|
> | N=16     | 29.76 | 30.43  | 27.97 | 30.84     | 27.69      | 31.56 |
> | N=32     | 29.63 | 30.43  | 27.91 | 30.77     | 27.87      | 31.62 |
> | N=48     | 29.49 | 30.36  | 28.00 | 30.70     | 27.62      | 31.61 |
>
> These results indicate that the method remains robust to the initial clustering setup. (The default number of first-layer clusters is N=64 in our main experiments)
> Please refer to section A.9 and Table 7 in our revised appendix.
>
>
> 2. More in-depth analysis of hyperparameters
>
> We thank the reviewer for highlighting this important point. We address each hyperparameter below:
> - Number of clustering layers (K): As shown in Appendix Table 6, we tested K = 2, 4, and 5 in addition to 1 and 3. Performance improves from K=1 to K=3, but further increasing K leads to slight drops due to increased parameter complexity within limited iterations. Based on this analysis, we choose K=3 as a trade-off that balances expressiveness with training efficiency.
> - Max scale constraint: The scale loss prevents Gaussians from becoming excessively large during deformation, which may otherwise introduce rendering artifacts. We found that setting a reasonable threshold (in our case, max_scale = 0.02) is sufficient. Our additional experiment below shows that disabling this constraint (setting max_scale to a very large value) will result in noticeable PSNR degradation. We have included this experiment in our revised appendix (section A.10 and Table 8).
>
> | Method  \ Scene       | Robot | Spring | Wheel | Pendulums | Robot-Task | Cloth |
> |----------------|-------|--------|-------|-----------|------------|-------|
> | max_scale=0.02     | 29.46 | 30.28 | 27.95 | 30.6  | 27.67 | 31.68 |
> | max_scale=2.0 | 15.86 | 19.86 | 22.42 | 18.86 | 18.15 | 21.53 |
>
> - Loss weights: We use fixed empirical weights: [0.19, 0.10, 0.19, 0.48, 0.05] for rigidity, isometry, rotation, scale, and RGB losses, respectively. We have included this in the revised paper (section 3.4).
>
> 3. Potential extension of the current pipeline
>
> We thank the reviewer for pointing out this important limitation. As also noted in Appendix A.12, our current method, while significantly reducing the number of training iterations (100 per frame), does not yet achieve real-time training or rendering. We agree that bridging this gap is a valuable and challenging direction. As part of our ongoing work, we are exploring strategies like GPU-specific kernel optimization to further improve efficiency.
>
> We have uploaded a revised main paper and appendix (included in the supplementary materials). For clarity, we have temporarily marked all revised texts in purple. The color changing will be removed and reverted to black text in the final version.

---

> > ### Comment · Reviewer_JCjr · 2025-05-05
> >
> > Thanks for the authors' reply. The revision has addressed most of my concerns. I also read other reviews and maintain my original opinion.

---

### Comment · Editors_In_Chief · 2025-07-04
**Post-Publication Edit**

On July 4, 2025, at the request of the authors, the Editors-in-Chief uploaded a new camera ready version. This version includes acknowledgments and updates some details in the references.

---

### Decision · Action_Editor_EdEC · 2025-05-29

**Recommendation:** Accept as is

**Comment:**

The idea of this work is interesting. Also, the contributions touch on key topics—efficient optimization, dynamic scene modeling, and self-supervision—that are squarely in TMLR’s scope.
The paper presents a solid contribution to 4D novel view synthesis with comprehensive experiments. The AE believes there are many potential applications in the presented paper.

This submission satisfies all the TMLR criteria on novelty, contribution, findings, and broad interests with high quality.

**Audience:**

Many readers of TMLR would find these results compelling. Also, the findings of revealing a self-supervised segmentation are interesting.
As this method could be leveraged in the robotics field (shown in figures), some robot-/computer vision-related audiences might be interested as well (broad interests).

**Claims And Evidence:**

This submission presents a structural, cascaded optimization framework for dynamic 4D view synthesis that leverages the internal structure of 3D Gaussian Splatting to dramatically accelerate convergence. It hierarchically decomposes the scene into coarse rigid motions and then refines them through finer, cluster-based adjustments.

Experiments show that SCas4D matches or exceeds the rendering quality of prior methods (e.g., Dynamic3DGS) while using only a fraction (∼1/20th) of their training iterations.

Comprehensive experiments verify the effectiveness of the proposed method well.